

# Utilising one minute and four minute recovery when employing the resistance training contrast method does not negatively affect subsequent jump performance in the presence of concurrent training

Dean Ritchie[1,2], Justin W.L. Keogh[1,3,4,5], Peter Reaburn[1] and Jonathan D. Bartlett[1,6]

[1] Faculty of Health Science and Medicine, Bond University, Gold Coast, QLD, Australia
[2] Gold Coast Suns FC, Gold Coast, QLD, Australia
[3] Faculty of Science, Health, Education and Engineering, University of the Sunshine Coast, Sunshine Coast, QLD, Australia
[4] Kasturba Medical College, Manipal Academy of Higher Education, Karnataka, India
[5] Sports Performance Research Centre New Zealand, Auckland University of Technology, Auckland, New Zealand
[6] Institute for Health and Sport, Victoria University, Melbourne, VIC, Australia

Corresponding author
Dean Ritchie,
dean.ritchie@goldcoastfc.com.au

## ABSTRACT

**Background:** Little is known about contrast training and post-activation performance enhancement (PAPE) in a same day concurrent training model. The aim of the current study was to examine the use of two short duration (1-min and 4-min) recovery periods on drop jump performance in same day concurrently trained athletes.

**Methods:** Ten professional Australian Rules footballers (age, 20.6 ± 1.9 yr; height, 184.8 ± 6.9 cm; body mass, 85.8 ± 8.4 kg) completed two resistance training sessions with different PAPE recovery durations; 1-min and 4-min, 1 h following a field-based endurance session. Baseline (pre) drop jumps were compared to post-test maximal drop jumps, performed after each set of three squats (where each participant was encouraged to lift as heavy as they could), to determine changes between 1-min and 4-min recovery periods. Data were analysed by fitting a mixed model (significance was set at $P \leq 0.05$). Corrected Hedges' g standardised effect sizes ±95% confidence limits were calculated using group means ± SDs.

**Results:** There were no significant differences between baseline and experimental sets 1, 2 and 3 for reactive strength index (RSI), flight time, and total and relative impulse for either recovery duration. However, for contact time, 1-min baseline was significantly different from set 2 (mean difference; 95% CI [0.029; 0.000–0.057 s], $P = 0.047$, ES; 95% CI [−0.27; −1.20 to 0.66]). For RSI and flight time, 1-min was significantly higher than 4-min (RSI: 0.367; 0.091 to 0.642, $P = 0.010$, ES; 95% CI [0.52; −0.37 to 1.42]; flight time: 0.033; 0.003 to 0.063 s, $P = 0.027$, ES; 95% CI [0.86; −0.06 to 1.78]).

**Discussion:** Short recovery periods of 1-min may be a time-efficient form of prescribing strength-power exercise in contrast loading schemes. Longer recovery periods do not appear to benefit immediate, subsequent performance.

# INTRODUCTION

Within the strength and conditioning literature, two commonly used models exist to periodise within-session resistance training; the traditional and contrast training methods. The traditional training approach structures training in accordance to the estimated cost of fatigue whereby plyometric, power and compound exercises are completed at the beginning of the resistance training session (*Kraemer et al., 2002, 2009*). The second method is contrast training which typically involves the performance of 3–5 repetitions with high loads of compound exercises, for example squats or power cleans, shortly followed by a lighter power or plyometric exercise (*Maio Alves et al., 2010*). This contrast mode of resistance training has become popular within team sport settings (*Comyns et al., 2007*; *Argus et al., 2012*; *Mola, Bruce-Low & Burnet, 2014*).

In the context of high-performance team sports, drop jumps are commonly utilised when executing a number of key technical skills that directly contribute to sport performance. While jump height has typically been the primary outcome assessed in jumping studies, ground contact time may also be important, as the same jump height achieved by a shorter ground contact time leads to a more efficient jump strategy. Reactive strength index (RSI) takes into account both jump height and ground contact time from drop jumps and is calculated as jump height divided by ground contact time (*Struzik et al., 2016*). Consequently, any contrast training related chronic improvements in an athlete's RSI and thus drop jump ability may improve the performance of many key technical skills and competitive match play.

The benefit of using the contrast training method reflects the ability to utilise post-activation potentiation (PAP) across multiple training sessions, thereby increasing the potential for positive adaptations in the long-term. PAP is the acute enhancement of force or muscle twitch contraction after a previous conditioning contraction or maximal voluntary contraction (*Gossen & Sale, 2000*; *Hamada et al., 1985*; *Tillin & Bishop, 2009*). Whilst PAP is typically associated with enhancements in muscle twitch properties, recent research indicates that post-activation performance enhancement (PAPE) is a more appropriate term in relation to contrast training as it reflects the ability to enhance voluntary force production (*Blazevich & Babault, 2019*).

Post-activation performance enhancement is utilised on the premise that the preceding strength-based movement may result in a complementary improvement in performance of the following jump or plyometric movement (*Tillin & Bishop, 2009*; *Gouvêa et al., 2013*). PAPE is generally elicited when a small number of dynamic repetitions (≤5) are

performed as explosively as possible with loads of 80–100% of repetition maximum (*Hamada et al., 1985*; *Tillin & Bishop, 2009*). Whilst the mechanistic underpinnings of PAP reflect two major pathways; the phosphorylation of myosin regulatory light chains (*Tillin & Bishop, 2009*; *Sweeney & Stull, 1990*; *Szczesna, 2003*; *Hodgson, Docherty & Robbins, 2005*; *Baudry & Duchateau, 1985*) and alterations in neural stimulation (*Tillin & Bishop, 2009*), PAPE mechanisms are more related to changes in muscle temperature and alterations to muscle force; effects purported to be driven by fluid changes to the working musculature (*Blazevich & Babault, 2019*).

Of specific relevance to the planning, prescription and organisation of resistance training in team sports, the rest interval between the conditioning contraction and subsequent plyometric or power exercise has been previously reported to affect the PAPE effect (*Gouvêa et al., 2013*). A meta-analysis has indicated that 8 to 12 min of recovery after the conditioning contraction is required to induce the greatest PAPE response (*Gouvêa et al., 2013*). However, in the context of team sport training, where large numbers of athletes train simultaneously within scheduled time blocks, the elongated recovery period of 8–12 min is impractical. In team sport athletes, a positive PAPE response has been demonstrated when utilising short (≤4 min) recovery periods. For example, *Mitchell & Sale (2011)* demonstrated in resistance-trained university rugby union players a 2.9% increase in CMJ height 4 min after 5-RM back squat with a self-selected load. The variations in the optimal recovery period to maximise the PAPE response may reflect variations in the PAPE protocol (e.g. intensity of the conditioning stimulus and the magnitude and time course of resulting fatigue) and the characteristics (e.g. strength and fatigability) of the participants (*Gouvêa et al., 2013*; *Tillin & Bishop, 2009*). This highlights the reciprocal relationship between PAPE and fatigue as evidenced by varying results in performance outcomes (*Tillin & Bishop, 2009*). The PAPE recovery period presents as a challenge, particularly in-season, since limited time is allocated to resistance-training (*Gamble, 2006*; *Ritchie et al., 2016*). Thus, a shorter recovery duration between contraction stimulus and subsequent plyometric/jump exercise is highly appealing as it represents a more ecologically valid training methodology for use in high-performance sport.

In further considering the ecological relevance of PAPE duration time and its increasing utilisation within team sport settings, no previous PAPE related research has been conducted in high level athletes performing same day concurrent training (CT) (*Comyns et al., 2007*, *Mola, Bruce-Low & Burnet, 2014*). Indeed, given that concurrently-trained athletes typically, by way of their training history, have high strength levels and are generally fatigue-resistant (*Ritchie et al., 2016*; *Bilsborough et al., 2015*), they may be able to demonstrate significant PAPE effects with shorter rest periods than that commonly recommended in the strength and conditioning literature (*Gouvêa et al., 2013*; *Mitchell & Sale, 2011*). Moreover, the training phase (i.e. pre- vs in-season) (*Ritchie et al., 2016*) and distribution of load (*Juhari et al., 2018*) across a week are altered highlighting why PAPE recovery time course may be important in relation to resistance training organisation. Accordingly, the aim of the current study was to examine the effect of short duration contrast training recovery time course (1-min) vs longer duration recovery time course (4-min) on drop jump (DJ) performance in the presence of same day

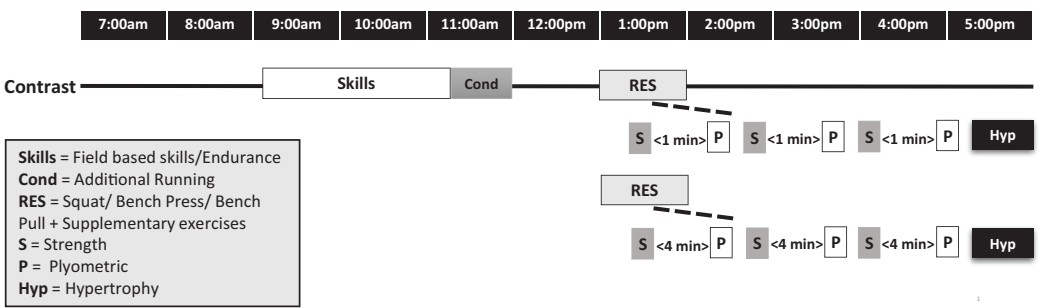

**Figure 1 Timeline shows team-based skills/endurance session in the morning followed by resistance training in the afternoon with either 1-min or 4-min recovery prescribed within contrast training on separate training days.**

concurrent training. The authors hypothesised that both short duration protocols would not attenuate subsequent DJ perfromance and will provide an ecologically valid recovery time course for PAPE in same day CT athletes.

# MATERIALS AND METHODS

In a randomised crossover design, participants completed two resistance training (RT) sessions with different PAPE recovery durations; 1-min and 4-min, following a field-based endurance session (endurance session external load matched between 1-min and 4-min protocols). Briefly, participants undertook a standardised same-day CT protocol where endurance training (predominately skill-based drills with the addition of top-up running) was completed in the morning followed by a RT session in the afternoon (Fig. 1) (*Ritchie et al., 2020*). All participants were provided 1 h of recovery between the endurance and RT training sessions and 48 h was allocated between the contrast training testing session and previous training session. Three baseline drop jumps (DJ) were performed after a standardised warm-up, followed by 3 sets of 3 squats where each participant was encouraged to lift as heavy as possible. DJ were performed with either 1-min or 4-min recovery after each set of squats. All participants had previously participated in and were familiar with heavy barbell box squatting and plyometrics movements where a range of force plate measured training metrics were assessed.

Ten professional Australian rules football (AF) athletes (mean ± SD: age, 20.6 ± 1.9 yr; height, 184.8 ± 6.9 cm; body mass, 85.8 ± 8.4 kg; Box Squat 1RM, 150 ± 16 kg; elite training age, 3.0 ± 1.2 yr) from the same Australian Football League (AFL) club participated in this in-season study. The participants competed in the national AFL competition with each providing written informed consent. All participants were required to be free from any injury or medical conditions throughout the data collection period that may affect their performance or endanger their health. If players suffered an injury, defined as pain resulting in modified load, data was excluded from final analysis. The project was approved by the University's Human Research Ethics Committee (DR03167).

Participants arrived at the training facility between 7:00 and 8:00 am before completing individual preparation for a team-based skills/endurance session. External training load of the field-based outdoor sessions (field-based skills/endurance) was monitored via

*Catapult S5 OptimEye* (Catapult Innovations, Docklands, VIC, Australia) global position system (GPS). Training metrics obtained were Total Distance metres (m) (TD), Total High Speed Running (>15 km·h$^{-1}$) (m), TD above 75% of an individual's maximum velocity (m) (75%), TD above 85% of an individual's maximum velocity (m) (85%) and mean running speed (m·min$^{-1}$). Maximal velocities were obtained as previously described (*O'Connor et al., 2019*). Each player wore the same GPS device across both outdoor sessions to account for between unit errors (*Rampinini et al., 2011*) as per manufacturer instructions (*Ritchie et al., 2016*). The accuracy and reliability of 10 Hz GPS units for quantifying the movement demands of team sport athletes have been previously reported (*Jennings et al., 2010*).

After the completion of the field-based skills/endurance session, participants consumed a mixed meal containing variable amounts of carbohydrate and protein, targeted towards each participants' own personal body composition and training goals. This was followed by passive rest until commencement of the RT session. At the beginning of the subsequent RT session, participants completed a standardised warm up which included bodyweight squats, mini-band lateral walks and pogo jumps followed by three submaximal DJ's. The participants were then given 2 min recovery before their baseline DJ were recorded. Participants were instructed to place hands on hips and to hold them there throughout the jump. Participants were then instructed to step off the box (30 cm) and land with two feet on a force place (400 series force plate; Fitness Technology, Adelaide, SA, Australia) simultaneously ensuring a short ground contact time and maximal rebound jump height with the aim to jump as high as possible. No knee bend was allowed during flight phase. If knee bend was observed, the jump was repeated. This type of DJ has previously been referred to as a bounce drop jump (*Struzik et al., 2016*). After three baseline jumps were completed, 2 min of recovery time was allowed then participants commenced the contrast training protocol. Table 1 displays the Intraclass correlation coefficients (ICC) and coefficient of variation (CV) for baseline DJ outcome measures.

Participants were instructed to complete a Barbell box squat utilising a self-selected 'heavy' load with participants instructed to lift as much as possible for the prescribed sets and reps. Three repetitions of the Barbell box squat (74% ± 9% of 1-RM) followed by a 1-min or 4-min rest period between each squat set before performing three maximal DJ repetitions. Force plate variables of contact time (CT) (s), flight time (FT) (s), impulse (N·s), relative impulse (N·s·kg$^{-1}$) and reactive strength index (RSI) (RSI = flight time (s)/ contact time (s)) (*Bosco, Luhtanen & Komi, 1983*; *Young, Wilson & Byrne, 1999*) were collected and recorded via the associated computer software (Ballistic Measurement System; Fitness Technology, Adelaide, SA, Australia). These methods have been previously proven as reliable and valid measures of assessing changes in lower body power (*Markovic et al., 2004*; *McGuigan et al., 2006*; *Cormack et al., 2008*; *Buckthorpe, Morris & Folland, 2012*).

Prior to the analysis of outcome measures, Shapiro–Wilk tests for normality and lognormality were demonstrated for the preceding sport specific running loads and PAPE squat conditioning stimulus. All outcome measures for DJ performance including modified RSI, impulse, relative impulse, flight time and contact time are presented as mean ± standard deviation. Paired *t*-test and Wilcoxon tests were utilised to assess significant differences between individual GPS training metrics (TD, high-speed running,
**Table 1 Intraclass Correlation (ICC) and Coefficient of Variation (CV) for baseline drop jumps.**

|  | ICC | CV (%) |
|---|---|---|
| RSI (reactive strength index) | 0.85 | 14.5 |
| Impulse (N·s) | 0.83 | 23.0 |
| Relative Impulse (N·s·kg$^{-1}$) | 0.86 | 23.0 |
| Flight Time (FT) (s) | 0.54 | 7.1 |
| Contact Time (CT) (s) | 0.86 | 12.9 |

**Table 2 Comparisons of 1 min vs 4 min protocols for preceding sports specific running loads.**

|  | Total distance (m) | | High speed running (>15 km/h) | | Distance covered >75% (m) | | Distance covered >85% (m) | | M/min | |
|---|---|---|---|---|---|---|---|---|---|---|
|  | 1 min | 4 min | 1 min | 4 min | 1 min | 4 min | 1 min | 4 min | 1 min | 4 min |
| Mean (SD) | 7,912 (1,258) | 7,805 (1,299) | 2,338 (772) | 2,280 (829) | 138 (80.4) | 159 (113) | 34.5 (26.7) | 42.9 (45.2) | 99 (13.7) | 96 (15.8) |
| *P* value | 0.724 | | 0.800 | | 0.695 | | 0.882 | | 0.695 | |
| ES (95% CI) | 0.08 [−0.80 to 0.96] | | 0.07 [−0.81 to 0.95] | | −0.21 [−1.08 to 0.67] | | −0.22 [−1.10 to 0.66] | | 0.19 [−0.68 to 1.07] | |

**Table 3 Comparisons of 1 min vs 4 min protocols for total tonnage of preceding conditioning stimulus (box squat).**

|  | Set 1 (kg) | | Set 2 (kg) | | Set 3 (kg) | |
|---|---|---|---|---|---|---|
|  | 1 min | 4 min | 1 min | 4 min | 1 min | 4 min |
| Mean (SD) | 299 (48.2) | 291 (47.0) | 318 (49.9) | 318 (45.2) | 327 (56.0) | 327 (42.9) |
| *P* value | 0.655 | | >0.999 | | >0.999 | |
| ES (95% CI) | 0.16 [−0.72 to 1.04] | | 0.00 [−0.88 to 0.88] | | 0.00 [−0.88 to 0.88] | |

distance covered >75% and 85% maximal velocity and m/min) and PAPE squat tonnage between the 1-min and 4-min protocols. The subsequent analyses were carried out separately for each dependent variable (RSI, impulse, relative impulse, flight time and contact time) with 3 comparisons per family (sets). Due to technical (a malfunction of the force plate software) constraints there were 10 missing values across the study of random participants. Therefore, data were analysed by fitting a mixed model, rather than by repeated measures ANOVA. Significance was set at $P \leq 0.05$. All data were analysed using (GraphPad Prism Version 8.04.1; GraphPad Software, La Jolla, CA, USA). Subsequently, bias corrected Hedges' g standardised effect sizes ±95% confidence limits were calculated using group means ± SDs via a customised Excel spreadsheet. Modified Cohens' d thresholds of small (0.2), moderate (0.5), and large (>0.8) were used to determine the magnitude of effect (*Cohen, 1988*).

## RESULTS

Preceding field-based skills/endurance (sport specific) same day running loads that precede the PAPE protocols are displayed in Table 2 with PAPE squat conditioning stimulus tonnage shown in Table 3. There were no significant differences in either

sport-specific running loads or PAPE conditioning stimulus load between the 1-min and 4-min protocols.

There were no significant difference between baseline 1-min and 4-min conditions for any of the jump performance variables (Fig. 2). Within condition analyses between baseline and experimental sets 1, 2 and 3 revealed no significant difference for RSI, flight time, total and relative impulse (Figs. 3 and 4). However, for contact time, 1-min baseline was different from set 2 only (mean difference; 95% CI [0.029; 0.000–0.057 s], $P = 0.047$, ES; 95% CI [−0.27; −1.20 to 0.66]) with no significant differences observed for 4-min.

Between condition set comparisons for both conditions are summarised in Fig. 5. For RSI, there was no difference between conditions for set 1 (mean difference; 95% CI [0.200; −0.214 to 0.612], $P = 0.472$, ES; 95% CI [0.26; −0.62 to 1.14]) and set 2 (0.309; −0.114 to 0.733, $P = 0.162$, ES; 95% CI [0.45; −0.48 to 1.39]). However, for set 3, RSI was significantly higher for the 1-min than 4-min condition (0.367; 0.091 to 0.642, $P = 0.010$, ES; 95% CI [0.52; −0.37 to 1.42]) (Fig. 5A).

Consistent with RSI, there was no difference between conditions for flight time for set 1 (0.027; −0.026 to 0.080 s, $P = 0.435$, ES; 95% CI [0.62; −0.28 to 1.52]) and set 2 (0.014; −0.048 to 0.078 s, $P = 0.870$, ES; 95% CI [0.34; −0.59 to 1.27]). However, for set 3, flight time was greater for the 1-min than 4-min condition (0.033; 0.003 to 0.063 s, $P = 0.027$, ES; 95% CI [0.86; −0.06 to 1.78]) (Fig. 5D).

For absolute impulse (N·s), there was no difference between conditions for set 1 (373; −2,932 to 3,678 N·s, $P = 0.984$, ES; 95% CI [0.07; −0.81 to 0.94]), set 2 (2,348; −1,763 to 6,459 N·s, $P = 0.314$, ES; 95% CI [0.37; −0.56 to 1.30]) or set 3 (2,744; −255 to 5,744 N·s, $P = 0.074$, ES; 95% CI [0.50; −0.39 to 1.39]) (Fig. 5B). Similarly, relative impulse (N·s·kg$^{-1}$) showed no difference between conditions (Set 1: 6.54; −31.47 to 44.56 N·s·kg$^{-1}$, $P = 0.948$, ES; 95% CI [0.08; −0.79 to 0.96]; Set 2: 34.57; −17.72 to 86.86 N·s·kg$^{-1}$, $P = 0.217$, ES; 95% CI [0.42; −0.51 to 1.36]; Set 3: 33.92; −5.58 to 73.42 N·s·kg$^{-1}$, $P = 0.097$, ES; 95% CI [0.45; −0.44 to 1.33]) (Fig. 5C).

For contact time there was no difference between conditions for any of the 3 sets (Set 1: 0.003; −0.034 to 0.040 s, $P = 0.992$, ES; 95% CI [0.06; −0.82 to 0.94]; Set 2: −0.014; −0.053 to 0.024 s, $P = 0.622$, −ES; 95% CI [0.27; −1.20 to 0.66]; Set 3: −0.017; −0.038 to 0.004 s, $P = 0.138$, ES; 95% CI [−0.26; −1.14 to 0.62]) (Fig. 5E).

## DISCUSSION

The major finding of the current study was that 1-min and 4-min had no negative effect on subsequent drop jump performance compared to respective baselines following same day skills-specific endurance training. Furthermore, when comparing 1-min vs 4-min between sets, both RSI and flight time significantly improved in set 3 with 1-min compared to 4-min protocol. Taken together, these data have implications for the organisation and planning of contrast resistance training in team sport settings where many athletes have relatively limited time to perform resistance training. Specifically, the results suggest that concurrent trained athletes with a strong aerobic/anaerobic conditioning base may be able to utilise shorter RT rest periods than is currently recommended for most other individuals.

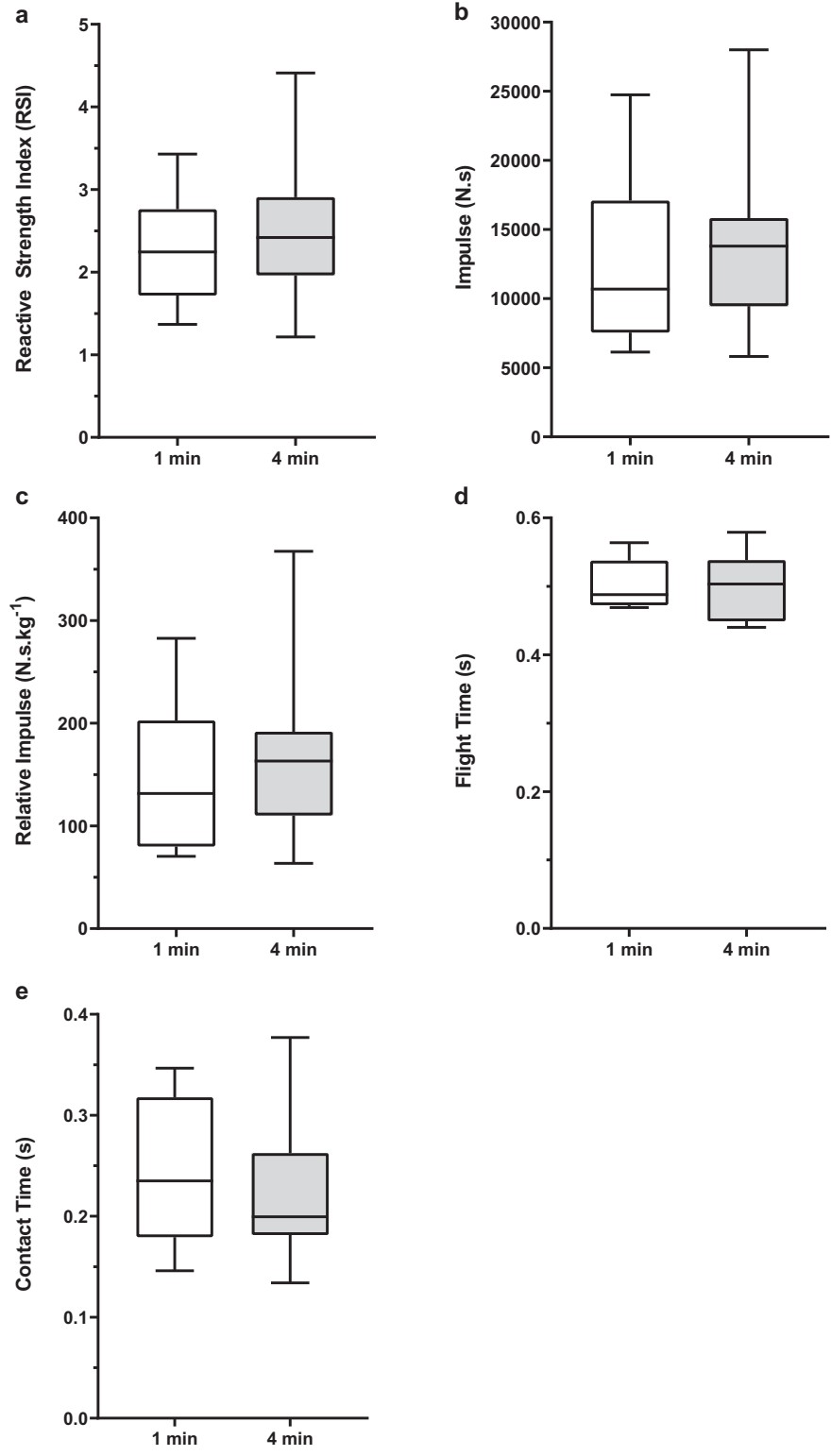

**Figure 2 A box and whisker plot showing baseline drop jump performance for 1-min and 4-min protocols.** (A) RSI, (B) Impulse, (C) Reactive Impulse, (D) Flight Time and (E) Contact Time. The box represents 25th and 75th percentiles and the bars represent minimum and maximum values.

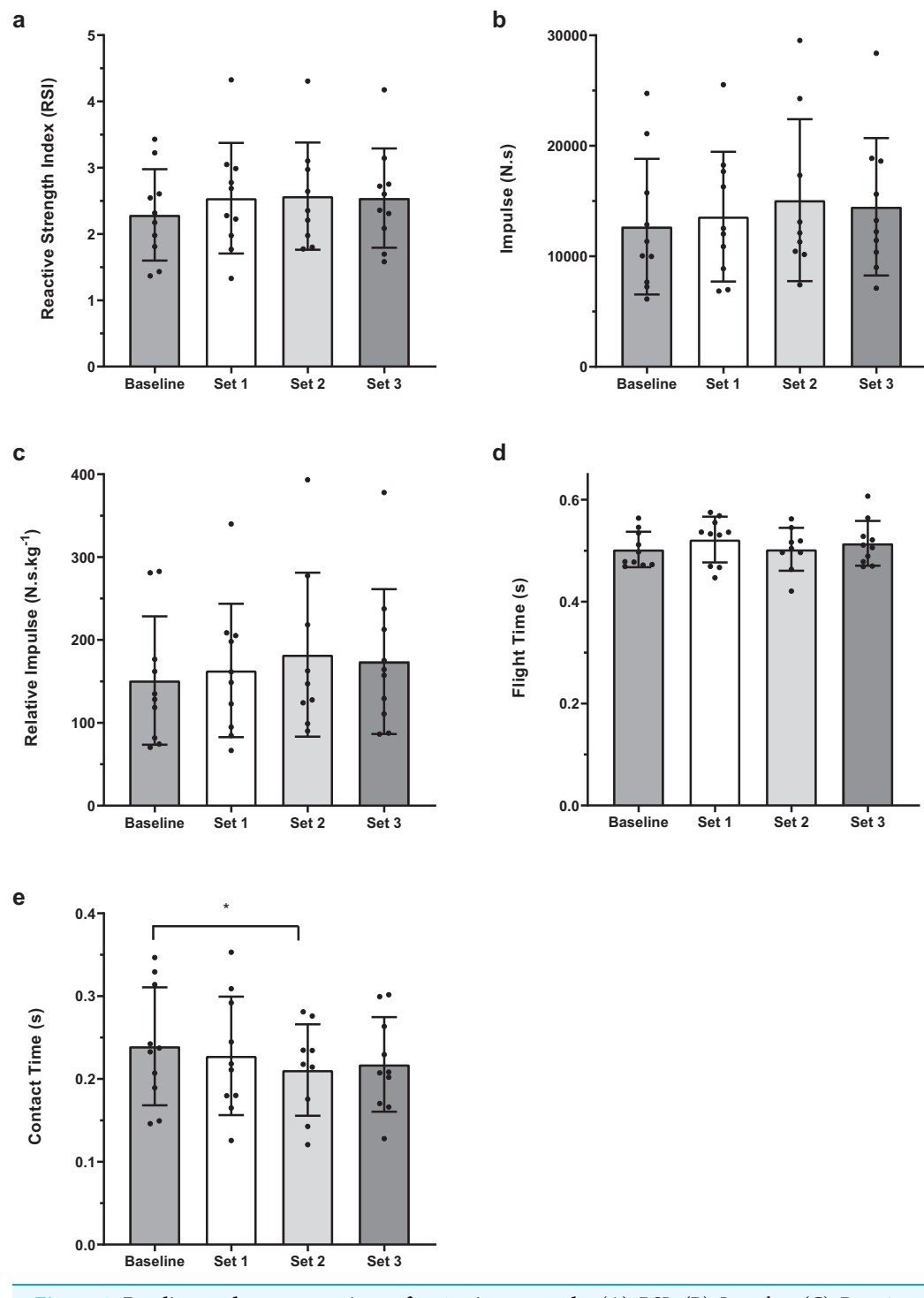

**Figure 3 Baseline and set comparisons for 1-min protocols.** (A) RSI, (B) Impulse, (C) Reactive Impulse, (D) Flight Time and (E) Contact Time. Bars represent mean and SD values, with individual data points plotted. An asterisk (*) indicates *P* value < 0.05.

The benefit of using CT reflects the ability to utilise preceding strength-based movement to improve performance of the subsequent jump or plyometric movement (*Tillin & Bishop, 2009*). In the current study no detrimental effect of utilising short 1- and 4-min

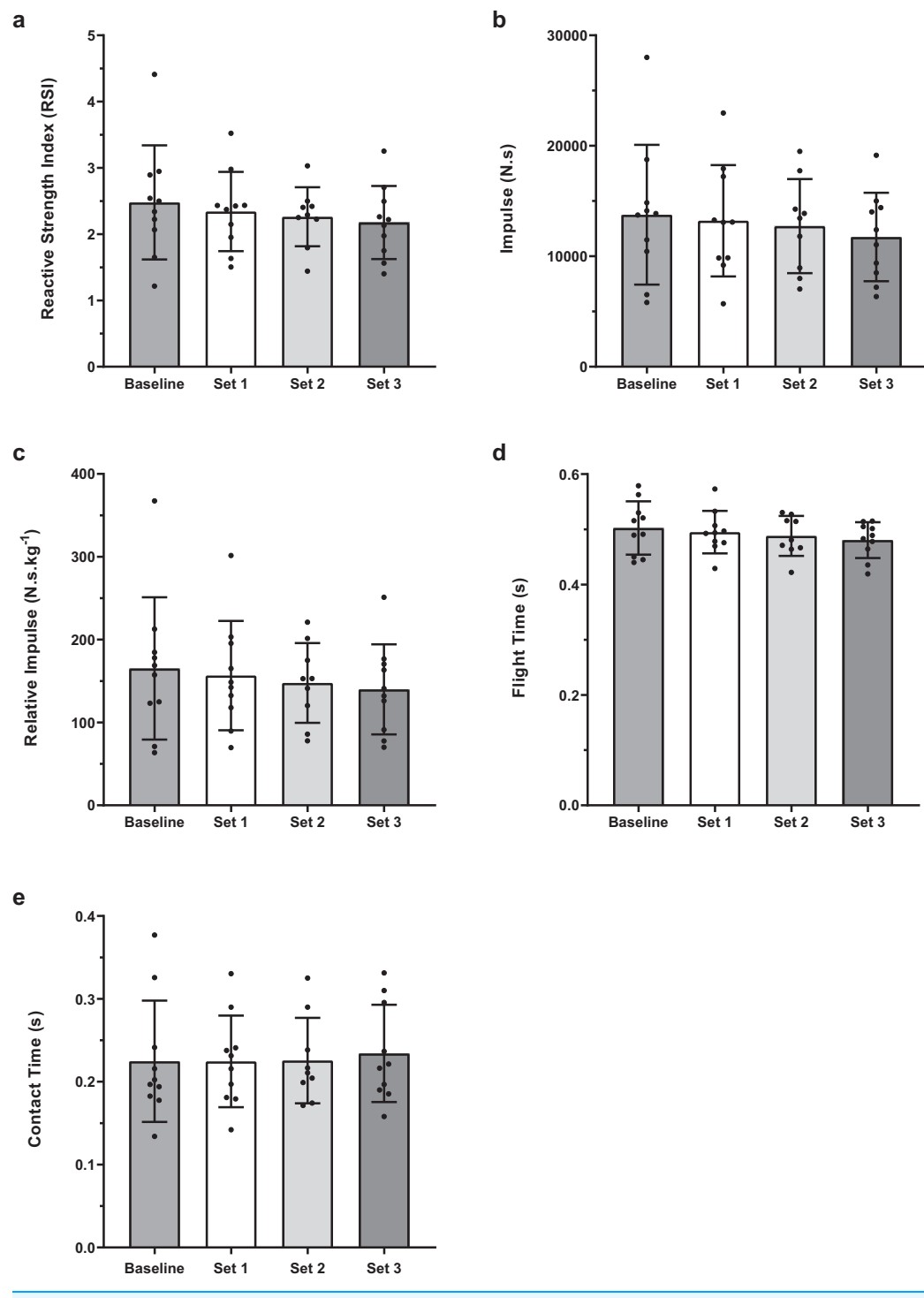

**Figure 4 Baseline and set comparisons for 4-min protocols.** (A) RSI, (B) Impulse, (C) Reactive Impulse, (D) Flight Time and (E) Contact Time. Bars represent mean and SD values, with individual data points plotted. An asterisk (*) indicates $P$ value < 0.05.

recovery durations was observed on subsequent drop jump performance following field-based skills/endurance running loads. This finding contributes to the current contrast training research literature, where a previous meta-analysis suggest that 8–12 min recovery

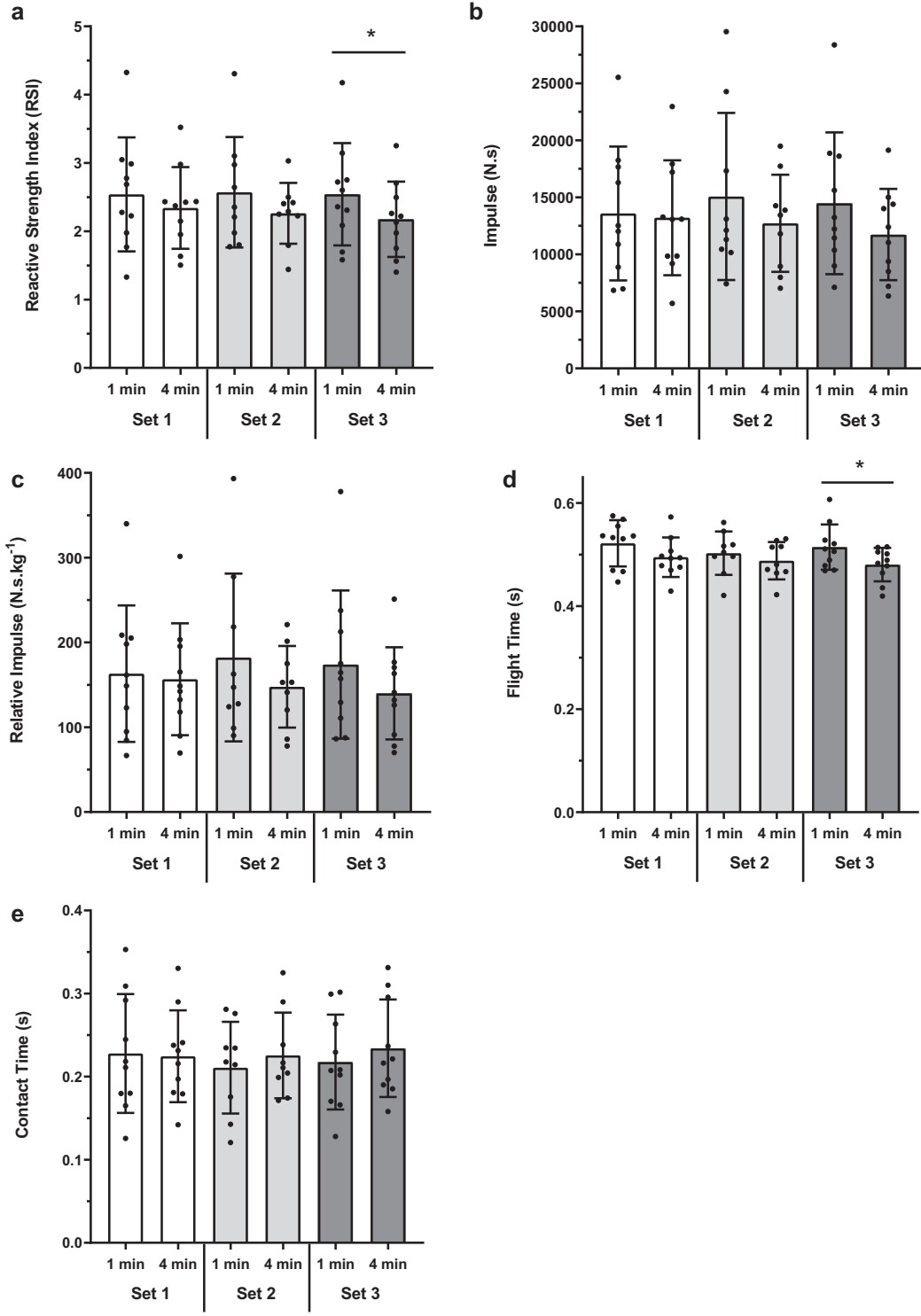

**Figure 5 Between condition set comparisons for 1-min and 4-min protocols.** (A) RSI, (B) Impulse, (C) Reactive Impulse, (D) Flight Time and (E) Contact Time. Bars represent mean and SD values, with individual data points plotted. An asterisk (*) indicates $P$ value < 0.05.

after the conditioning contraction produces the greatest PAPE effect (*Gouvêa et al., 2013*). Importantly, individual studies presented in the meta-analysis indicate that the resultant PAPE may be observed anywhere from between 4 and 24 min (*Gouvêa et al., 2013*). However, due to the complex nature of team sports weekly dense schedules (*Ritchie et al., 2016*) where several other aspects of coaching, recovery and mental well-being are required, if employing the recommended 8–12 min recovery period (*Gouvêa et al., 2013*) between 3 and 4 sets of heavy back squat exercise and subsequent plyometric exercise, the time to complete these exercises would be >40 min. This length of recovery period provides limited time to complete supplementary exercises within a typical resistance training session of 60 min and is impractical in the applied high-performance team training setting.

The utility of the DJ in the current study and not countermovement jump or squat jump can be explained by the importance of high levels of reactive strength produced over short contact times required as part of an efficient DJ (*Comyns et al., 2007*; *Argus et al., 2012*; *French, Kraemer & Cooke, 2003*), and the direct relevance of the vertical jump to Australian Rules football specific actions. The short contact is represented by a fast stretch shortening cycle (SSC) which is defined as ground reaction forces ranging from 100 to 250 ms (*Schmidtbleicher, 1986*). The present results showed an increase in RSI and FT in set 3 of 1-min compared to 4-min. These findings are important as RSI, expressed as the relationship between flight and contact time, represents explosive strength and the ability to develop maximal force in minimal time. Indeed, the ability to perform explosive actions utilising the SSC is a requisite for most sports (*Ebben & Petushek, 2010*). Given that high-intensity intermittent activity requires accelerations, change of direction and decelerations to be performed with maximal force almost instantaneously through competition (*Ritchie et al., 2016*; *Jennings et al., 2010*), RSI is a particularly important attribute for team sport athletes. Furthermore, it has been shown that in athletes who have suffered significant lower limb injures that the ability to produce force quickly is often attenuated in both the acute rehab and post return to play period (*Read et al., 2020*; *Jordan et al., 2020*). Thus, the current results present a useful method of within resistance training session organisation that can augment RSI for concurrently trained athletes such as team sport athletes.

The relationship between PAPE and fatigue is important when optimising the potentiation response. AF and more broadly team sport athletes possess a high aerobic capacity, and as team sports require repeated high-intensity actions throughout training and matches this may explain the present results showing the ability to recover adequately when utilising short duration recovery periods. The present results show no detrimental effect of short duration (<4 min) recovery periods on subsequent kinetic or kinematic variables of jump performance and that 1-min was more beneficial than 4-min. Previous research utilising 11 university rugby union players tested the effect of a 5-RM back squat on PAPE and subsequent CMJ performance with results showing a 2.9% increase in CMJ height 4 min after a 5-RM back squat (*Mitchell & Sale, 2011*). This finding may be explained by research showing twitch potentiation to be greatest immediately following a prior conditioning contraction (*Hamada et al., 1985*;

*Folland, Wakamatsu & Fimland, 2008*). However, although twitch potentiation may be the greatest immediately following a prior conditioning contraction, there also exists a high level of fatigue that could limit subsequent maximal performance (*Comyns et al., 2007*; *Pearson & Hussain, 2014*). For example *Pearson & Hussain (2014)* observed an increase in twitch torque of the knee extensors but no significant effect on CMJ jump height, jump power, rate of force development or take-off velocity 4 min post either a 3, 5 or 7 s isometric half squat MVC in recreationally trained men. The researchers suggested that PAPE was repressed by fatigue in the other musculature used during the conditioning stimulus, where twitch torque was only measured in the quadriceps (*Pearson & Hussain, 2014*). This may suggest a reciprocal relationship between fatigue and potentiation whereby if fatigue is favoured then a decrease in performance can be expected. Consequently, if potentiation is more pronounced, then an increase in performance might be expected (*Rassier & Macintosh, 2000*).

In the present study, the participants preceded the resistance training session by a sport-specific endurance training session typically completed as part of their normal training routines. The prior training suggests some residual fatigue upon commencing the subsequent resistance training session undertaken 1 h later. As such, participants were instructed to self-select their conditioning stimulus load while being told to lift as heavy as they could do safely. Participant's self-selected conditioning stimulus equated to 74% ± 9% of 1-RM which is less than the previous PAPE recommendations of 80% 1-RM (*Tillin & Bishop, 2009*). The acute strength 'state' of concurrent training athletes undertaking preceding high-intensity intermittent activities can fluctuate based on arousal, preceding activity, diet and sleep (*McBurnie et al., 2019*) inferring that the preceding sport-specific endurance session may influence subsequent resistance training performance. Consistent with this notion, it has been previously demonstrated that sport-specific endurance running loads negatively affect subsequent same day RT performance (*Ritchie et al., 2020*), suggesting that team sport athletes may not lift >80% 1-RM due to residual fatigue. In support of the efficacy of lower conditioning stimulus loads, previous research in national level Olympic lifters report increased CMJ performance following 'moderate' loads (45–75% RM) in a back squat (*Fukutani et al., 2014*). In addition, a meta-analysis indicates that moderate (60–84%) vs heavy loads (>85%) (ES: 1.06 vs 0.31), multiple set vs single set (ES: 0.66 vs 0.24) and athletic vs untrained cohorts (ES: 0.81 vs 0.14) result in an increase in power augmentation in subsequent potentiation tasks (*Wilson et al., 2013*). Greater RT experience results in increased neural firing, motor unit synchronisation (*Folland & Williams, 2007*) and elevated myosin regulatory light chain phosphorylation activity (*Sweeney & Stull, 1990*; *Hodgson, Docherty & Robbins, 2005*; *Baudry & Duchateau, 1985*). As such, it may be suggested that greater training experience facilitates the relationship between potentiation and fatigue, and the acute strength 'state' of the athlete can fluctuate, thus, current recommendations of >80% 1-RM for the conditioning contraction stimulus should only be considered a guideline.

Contrast training offers another method of within resistance training periodisation. However, little research has employed concurrently trained athletes as the experimental cohort (*Comyns et al., 2007*; *Mola, Bruce-Low & Burnet, 2014*) and no bounce drop jump

the PAPE response in a same day concurrent training model where the preceding field-based skills/endurance training load has been reported. The strength of the present study was that the preceding skills/endurance training load was matched between the 1-min and 4-min groups. Furthermore, the self-selected conditioning stimulus load was consistent between groups. In addition, the current collection period was undertaken during the in-season period where the focus is solely on being in peak condition for competition and subsequent recovery for the following game. This provides a high level of ecological validity as high performance athletes such as those involved with the present study will often self-select their individual resistance training loads based on their individual recovery status following the most recent competitive match. This meant that athletes lifted under the recommended 80% 1-RM previously reported in previous PAPE research, making comparisons with research utilising >90% 1-RM difficult (*Kilduff et al., 2008*; *Young, Jenner & Griffiths, 1998*; *Duthie, Young & Aitken, 2002*). Given this 'lower' conditioning stimulus load, it could be suggested that subsequent drop jump performance was compromised. However, drop jump performance following the conditioning stimulus was comparable to baseline meaning there was minimal detrimental effect of the lower prior PAPE stimulus.

Despite these novel findings, there are some inherent limitations that should be addressed. The current research utilised ten professional team sport athletes (Australian Rules Footballers) and while this was sufficient for the current analysis, larger sample sizes of athletes from Australian Rules Football and other team sports may increase the generalisability of these results. Furthermore, the absence of an 8–12 min recovery group for comparison as with current contrast training recommendations is a limitation. However, this is not feasible within the constraints of training design within the context of professional team sport athletes. In addition, Table 1 shows the ICC and CV for baseline drop jumps, while the ICC reported are classified as good reliability (*Koo & Li, 2016*), the observed CV are above the desired 5%. The higher than recommended CVs reflected the accumulated fatigue on the prior on field session, the relative complex nature of drop jumps compared to squat or countermovement jumps and the fact that a number of the variables for example RSI and impulse were calculated using DJ kinetic and temporal measures. However, the ecological validity of testing professional athletes in the field is a strength of the current study design.

## CONCLUSION

The present findings suggest the utilisation of PAPE within a same day concurrent training team sport model, where short duration recovery periods between conditioning stimulus and subsequent plyometric (jumping) tasks can be implemented without concern of negatively affecting the subsequent plyometric task performance. If the rest periods required for potentiation to exceed fatigue are too large, it is likely that strength and conditioning coaches working in high performance sporting organisations would have to reduce training volume or increase the duration of each resistance training session, both of which are not practical. As such, PAPE that utilises short recovery durations provides another within resistance training periodisation strategy for team sport athletes.

### Funding

The authors received no funding for this work.

### Competing Interests

Justin Keogh is an Academic Editor for PeerJ. Dean Ritchie is employed by the Gold Coast Suns FC.

### Author Contributions

- Dean Ritchie conceived and designed the experiments, performed the experiments, analysed the data, prepared figures and/or tables, authored or reviewed drafts of the paper, and approved the final draft.
- Justin W.L. Keogh conceived and designed the experiments, analysed the data, authored or reviewed drafts of the paper, and approved the final draft.
- Peter Reaburn conceived and designed the experiments, analysed the data, authored or reviewed drafts of the paper, and approved the final draft.
- Jonathan D. Bartlett conceived and designed the experiments, performed the experiments, analysed the data, prepared figures and/or tables, authored or reviewed drafts of the paper, and approved the final draft.

### Human Ethics

The following information was supplied relating to ethical approvals (i.e. approving body and any reference numbers):

The project was approved by the Bond University, Gold Coast Australia, Human Research Ethics Committee (Ethics reference number DR03167).

### Data Availability

The raw data and ICC and CV data are available in the Supplemental Files.

### Supplemental Information

Supplemental information for this article can be found online at http://dx.doi.org/10.7717/peerj.10031#supplemental-information.

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
