# Peer review of "Utilising one minute and four minute recovery when employing the resistance training contrast method does not negatively affect subsequent jump performance in the presence of concurrent training"

_PeerJ, doi:10.7717/peerj.10031_

## Round 0.1 · original submission · Major Revisions

Reviewers generally reported positive feedbak about the presented manuscript. However, several changes need to be implemented based on the specific suggestions provided by the reviewers.

Reviewer 1 ·

Basic reporting

No comment.

Experimental design

The technique of performing a drop jump can have a significant impact on the results of your work. Looking at the results, it can be assumed that it was bounce drop jump and not countermovement drop jump. See Struzik et al. 2016 for more details. The description of the jump should be more detailed. Alternatively, add to the limitations the fact that there was no detailed control of the jump technique.

Validity of the findings

No comment.

Additional comments

The article seems very interesting for the potential reader. Conclusions regarding a recovery time between exercises can be used in sports training.

Reviewer 2 ·

Basic reporting

Very well designed and written study. Literature is sufficient. Article structure is good, although I suggested minor improvements

Experimental design

Research question is well designed and supported. Authors studied appropriate sample of subjects and used appropriate methods.

Validity of the findings

Findings are valid and realistic, especially taking into account proper statistical methods and good interpretation.

Additional comments

Well done study. However, text is very dense and difficult to read. Also, PeerJ is e-journal and therefore there is no need for usage of "uncommon" abbreviations (i.e. ONEW, FOUR); use full words throughout the text (1-min-rest; 4-min-rest, etc.).
My main comment (major one) is related to presentation of the study. I'm not novice in science and still had to read experiment couple of time before understanding what authors did in their study. I strongly suggest (actually, I insist) on providing the Figure where experiment will be presented with differences between protocols, timing, etc.
Also, studies of such kind ask for a profound explanation of study limitations.
I would rather see last paragraph of the Discussion in the Conclusion section.

Reviewer 3 ·

Basic reporting

• Suggest the title may be re-considered to be more brief whilst still highlighting the purpose of the study
“The effects of short and long recovery periods during contrast resistance training in elite Australian Football”
• The use of the terminology “Post Activation Potentiation” should be reconsidered throughout the text. Contrast training (and recovery periods) can, and should be, be related to PAP in the introduction and discussion sections. However, this study is simply comparing 1min v 4min recovery periods contrast training. PAP does not need to form a key part of this manuscript.
• In addition to the above point, upon revising the use of the term, the authors should be clear in differentiating between “post activation potentiation” and “post activation performance enhancement (PAPE)” and address throughout. See: Blazevich et al. 2019 Post-activation Potentiation Versus Post-activation Performance Enhancement in Humans: Historical Perspective, Underlying Mechanisms, and Current Issues
• Abstract - see above regarding use of PAP - it doesn’t appear to be necessary in the abstract.
• Abstract - Methods - Last sentence can be changed to “Baseline (pre) drop jumps were compared to post-test drop jumps after each set of squats, to determine changes between 1min and 4min recovery periods”. Then include the statistical tests used.
• Abstract - Discussion - As mentioned, the PAP terminology is not appropriate here and needs re-wording. Findings are simply that “Contrast resistance training utilising 1min recovery durations were not detrimental to subsequent performance compared with 4min recovery”. These findings do not necessarily mean it is an ecologically valid method of periodization. We can’t assume from this data what might happen after longer recovery periods, or the long term adaptations. A more appropriate summary might be that short recovery periods of 1min may be time-efficient form of prescribing strength-power exercise. Longer recovery period do not appear to benefit immediate, subsequent performance and therefore may not be a time efficient method of training. This should be considered for the discussion section as well.
• Running head - As above, this is not an appropriate running head and doesn’t reflect the study. Please amend to something more reflective of the manuscript.

Line 55-56 - Wording makes it sounds like there are only two methods. Reword to “…two commonly used models to periodise resistance exercise within-session are the ‘traditional’ and ‘contrast’ training methods. The traditional training approach…”
Line 59 - Be more specific than “a few” even it is a repetition range supported by the relevant reference
Line 63-74 - It’s unclear why it is important to “utilise PAP” in this setting. I.e. why is it a benefit for an AFL player to utilise PAP? Can a link be made to the importance of improving jump height during the session? Or is it an effective method of developing long term strength-power qualities? This remains unclear to me throughout and needs to be substantiated if it is the path you want to take in this paper. An alternative path is to simply compare the two recovery durations, and then draw upon past PAP/PAPE literature to explain or discuss your findings.
Lines 86-90 - I think this makes a lot of sense and as above could be a better direction to take this paper. I.e. Recovery periods of 8-12 minutes have been suggested to maximise PAPE response, however are not appropriate in the applied environment. As such, the aim of this study was to compare the short term response following two smaller recovery periods of 1 and 4 minutes.
Intro - Why is RSI used? This probably warrants at least half a paragraph as the primary measure of performance.

Experimental design

Line 92 - I think it is important to state here that it is after other training methods rather than simply in the presence.
Line 106 - Was this a crossover study? I.e. All participants completed both 1min and 4min protocols? It seems so but a little unclear. If not, re-word this sentence to “… participants completed either 1min or 4min contrast training sessions”.
Line 106 - Would also be good to include a sentence or two on if and how training was standardised prior to this day of training and testing. E.g .48 hours of rest, when the last training session was completed.
Line 110 - Combine the next sentence here by including “1 hour later” after “RT session”.
Line 111 - How many baseline jumps?
Line 112 - How many repetitions?
Line 114 - Consider just using “1min” and “4min” here and throughout the text? Reads a little easier and doesn’t need to be simplified further.
Line 124 - The clubs committee, or the university?
Line 150 - Include how many jumps were used for baseline. Also include ICC and CV of the drop jump test in these athletes, to provide more context for results presented.
Line 168 - “Due to” instead of “Because of”
Line 168 - How many missing values were there?

Validity of the findings

Line 170 - I strongly encourage the authors to re-consider the use of “significance” thresholds and the terminology of “non-significant” and “significant” changes. Most statisticians now recommend simply reporting p values observed and effect sizes without categorizing at significant or non-significant. See following:
• Earnest et al. https://www.mdpi.com/2075-4663/6/4/139
• Hurbert et al. https://www.tandfonline.com/doi/full/10.1080/00031305.2018.1543616?fbclid=IwAR3WUKY_wcy_Uzx1uesINfmjOckY2T16ASpy9zH1XAQdI8_qlIJ1JgG5_iE
• Curran-Evertt et al. https://journals.physiology.org/doi/full/10.1152/advan.00054.2020?fbclid=IwAR17kNXspfXAtd_j9Z8Pm89-yi2p5iHxgnWGawf9uh8Lm2ak3iN3uZPULm0

Results section - It may be more insightful to include (at least in visualising in all figures) individual percentage changes of variables compared to baseline rather than presenting group means. This will help provide more context to the changes, or lack of changes reported.
Lines 178 - 185 - I like the inclusion of this data, certainly important to consider.
Figure 2 and 3 - Would it benefit the reader to overlay the results displayed in Figure 2 and 3? It may help to highlights the differences between the two protocols.

Additional comments

Thank you for the opportunity to review this manuscript. The study aimed to compare the use of a short recovery period (1min) and a longer recovery period (4min) when prescribing contrast strength-power training to elite Australian Football athletes.
The authors should be commended on writing this paper - studies are always difficult to complete in elite applied environments. I believe this study can contribute to the strength & conditioning knowledge, however, some general and specific changes (outlined below) need to be addressed before a decision can be made on this manuscript. Firstly and most importantly, the aims and reasoning behind the aims of the study, need to be clearly stated. The WHY of this study isn’t obvious as a reviewer. It is currently unclear why acute changes in RSI during training are important for athlete development / performance. It is much clearer why reducing rest periods would be beneficial for practitioners, particularly if there is no benefit of longer rest periods. This may be the direction you want to take with this study. I have outlined in further detail below.
Additionally, particularly given the small sample size, further work needs to be done to provide clarity over the reliability of the RSI testing measures (CV and ICC). Further, results should be presented as a percentage change in each individual.
Due to the required changes above and the specific changes below, I have not reviewed the discussion section. Upon resubmission, I would be happy to review the paper again.

·

Basic reporting

- The structure of the article conforms to the PeerJ standards and the raw data is well presented as part of this review.
- Clear and unambiguous English is used throughout. There are however quite a few instances where the author uses pronouns like "we" and "our" in the text which may detract from the professional scientific nature of the document.
- The structure of the document is sound with all the relevant information submitted for a thorough review.
- The document is self-contained with relevant results to the hypothesis.

Experimental design

- The authors have conducted a good, novel, research study which sought to validate current intra-session programming and periodisation.
- The research question is well defined, relevant, and meaningful to practitioners in the field.
- The study looks to have been conducted rigorously to a high technical and ethical standard.
- The methods require some clarification, as per the notes related to the methods section in the basic reporting section.

Validity of the findings

- The findings from the current study seem to be sound and serve to inform best practice for the strength and conditioning industry.
- The conclusion effectively summarises the key findings from the current study.

Additional comments

This article investigates a topical issue in strength and conditioning and seeks to bridge the gap between the current scientific understanding and the “real-world” application thereof. The outcomes from this study may be well received by practitioners, especially those working with high-intensity invasion sports athletes who participate in concurrent training.
- The article needs to be further proofread and written from a more scientific perspective. Throughout the document, the author uses personal pronouns like we (lines 102, 234, 291, 292, 298 and 326) and our (line 270) which may distract from the scientific nature of the document. An additional, very minor issue, is the amount of “filler information” within the document e.g. Line 76: “ To optimise the PAP response, a meta-analysis has indicated that 8 to 12 minutes of recovery after the conditioning contraction is required to induce the greatest PAP response (10).” This sentence may read better with the removal of the lead-in statement: “A meta-analysis has indicated that 8 to 12 minutes of recovery after the conditioning contraction is required to induce the greatest PAP response (10).”
- Ensure that a consistent term is used when referring to participants/athletes/subjects. In the current documents, the author refers to both participants and athletes. The same goes for “Australian Football” and “Australian Rules Football” which may be confusing for international readers.

Abstract:
- Add in information related to the rep schemes and potentially how this influenced load selection in the respective sets.
- Clarify that each set was followed by 3 maximal jumps. e.g. “Baseline drop jumps were collected, followed by 3 sets of 3 reps, during which athletes were encouraged to lift as heavy as possible. Following each set of squats, the athletes completed either the ONE or FOUR intervention followed by 3 maximal drop jumps.”

Introduction:
- There are a couple of relevant points introduced in the discussion section for the first time which should be added and covered in the introduction e.g. Lines 274-278. Defining/outlining the fitness and fatigue relationship, specific to PAP, in more depth in the introduction serves to provide better rationalisation of the importance of the research question. When the fatiguing effect of the squat priming stimulus is looked at in conjunction with the fatigue induced during the field-based session, the research question is better contextualised.
- Introducing information related to PAP interventions performed at lower percentages of 1RM e.g. plyometrics and Fukutani et al. (40), in conjunction with the “traditional” >80% 1RM protocol (Line 69), may serve to better lay the foundation for the outcomes of this study, where the average priming load was around 74% 1RM.
- It is advised that the author clearly state the rationale or performance/training benefits of using PAP interventions to potentiate plyometric or explosive exercises/activities. Why is this desirable in the training environment e.g. neurological benefits?
- Information in Lines 78-79 seems to be repeated in lines 86-88. Look to state this only once.
- References may be required for “strength levels” and “fatigue resistant”, Line 95, especially when looking at the relevance to PAP and the dissipation of fatigue. Tying these concepts together serves to back the hypothesis of the article.

Methods:
- It may be prudent to state the training age of the athletes. This may inform the training history and experience related to physiological, especially neurological training level and efficacy?
- Ensure all terms are written out in full prior to the use of the abbreviation e.g. Meter and m.
- Ensure that you provide a statement around session ONE and FOUR being load matched as per Lines 316/7, in the methods section. This will serve to strengthen the relevance of the outcomes of this study.
- Add in information related to the rep schemes and potentially how this influenced load selection in the respective sets.
- Include the information from Lines 250-253 into the methods section to justify the use of squat jump over the countermovement jump for plyometric assessment.

Results:
- As an aside… It would be interesting to see the individual responses to the different protocols. Are there grounds for individualised prescription of PAP rest prescription for athletes in the future?

Discussion:
- Ensure that all the points raised within the introduction and discussion sections are complementary, with the discussion building on key statements from the introduction. There are a couple of new concepts introduced in this discussion which need to be alluded to in the introduction e.g. the fitness: fatigue interaction related to PAP interventions.
- To strengthen this study, it may be prudent to state the resistance training age/experience level of the athletes. Especially when related to the neurological aspects mentioned in 306 - 307.
- Clarify when this study took place. Was it the pre-season or in-season?

---

## Round 0.2 · accepted · Accept

Reviewers and myself are happy with the changes implemented in the manuscript. Congratulatons for meeting the high standard publication of PeerJ.

Reviewer 1 ·

Basic reporting

The manuscript was revised my corrections request in the first round.

Experimental design

The manuscript was revised my corrections request in the first round.

Validity of the findings

The manuscript was revised my corrections request in the first round.

Additional comments

The manuscript was revised my corrections request in the first round.

Reviewer 2 ·

Basic reporting

Authors properly responded to my asks and amended the manuscript according to my suggestions. Thank you

Experimental design

Proper and well explained experimental design

Validity of the findings

Findings can be considered as valid

Additional comments

Thank you for amending the manuscript according to my comments and suggestions.

Reviewer 3 ·

Basic reporting

No comment

Experimental design

No comment

Validity of the findings

The inclusion of ICC and CV values, and discussion of these values greatly improved the conclusions of this paper. Considering the CV and ICC values, my only suggestion is that further comment on the use of the drop jump in future research may benefit researchers in this area. Particular when other options are available to assess force-time characteristics using the CMJ or SJ as mentioned in text.

Additional comments

Well done on putting together this re-submission. I believe the quality of this paper has been enhanced considerably.